# Structural basis of ion – substrate coupling in the Na+-dependent dicarboxylate transporter VcINDY

David B. Sauer[1,2,4], Jennifer J. Marden[1,2], Joseph C. Sudar [2], Jinmei Song[1,2], Christopher Mulligan [3✉] &
Da-Neng Wang [1,2✉]

The Na+-dependent dicarboxylate transporter from *Vibrio cholerae* (VcINDY) is a prototype
for the divalent anion sodium symporter (DASS) family. While the utilization of an electro-
chemical Na+ gradient to power substrate transport is well established for VcINDY, the
structural basis of this coupling between sodium and substrate binding is not currently
understood. Here, using a combination of cryo-EM structure determination, succinate binding
and site-directed cysteine alkylation assays, we demonstrate that the VcINDY protein
couples sodium- and substrate-binding via a previously unseen cooperative mechanism by
conformational selection. In the absence of sodium, substrate binding is abolished, with the
succinate binding regions exhibiting increased flexibility, including HP$_{in}$b, TM10b and the
substrate clamshell motifs. Upon sodium binding, these regions become structurally ordered
and create a proper binding site for the substrate. Taken together, these results provide
strong evidence that VcINDY's conformational selection mechanism is a result of the
sodium-dependent formation of the substrate binding site.

[1] Department of Cell Biology, New York University School of Medicine, New York, NY 10016, USA. [2] Skirball Institute of Biomolecular Medicine, New York
University School of Medicine, New York, NY 10016, USA. [3] School of Biosciences, University of Kent, Canterbury, Kent, UK. [4] Present address: Centre for
Medicines Discovery, Nuffield Department of Medicine, University of Oxford, Oxford, UK. ✉email: c.mulligan@kent.ac.uk; da-neng.wang@med.nyu.edu

VcINDY is a Na$^+$-dependent dicarboxylate transporter that imports TCA cycle intermediates across the inner membrane of *Vibrio cholerae*[1,2]. The detailed structural and mechanistic understanding of VcINDY[1–4] has made the protein a prototype of the divalent anion sodium symporter (DASS) family (Supplementary Fig. 1a, b)[5]. Within the human genome, the SLC13 genes encode for DASS members including the Na$^+$-dependent, citrate transporter (NaCT) and dicarboxylate transporters 1 and 3 (NaDC1 and NaDC3)[6]. Besides functioning as TCA cycle intermediates, DASS-imported substrates are central to a number of cellular processes. In bacteria C4-carboxylates can serve as the sole carbon source for growth[7], while imported citrate and tartrate are electron acceptors during fumarate respiration[8]. Citrate is also a precursor for both fatty acid biosynthesis and histone acetylation in mammals[9,10]. Dicarboxylates such as succinate and α-ketoglutarate act as signaling molecules that regulate the fate of naive embryonic stem cells and certain types of cancer cells[11,12]. As a result of these roles in regulating cellular di- and tricarboxylate levels, mutations in DASS transporters have dramatic physiological consequences. Deletion of bacterial DASS transporters can abolish growth on particular dicarboxylates[7,8]. Mutations in the human NaCT transporter cause SLC13A5-Epilepsy in newborns[13], whereas variants in the dicarboxylate transporter NaDC3 lead to acute reversible leukoencephalopathy[14]. In mice, knocking out NaCT results in protection from obesity and insulin resistance[15]. Such roles of SLC13 proteins in cell metabolism have made them attractive targets for the treatment against obesity, diabetes, cancer and epilepsy[16–18]. Therefore, mechanistic characterization of the prototype transporter VcINDY will help us to better understand the transport mechanism of the entire DASS family, including the human di- and tricarboxylate transporters.

The VcINDY protein is a homodimer consisting of a scaffold domain and a transport domain (Supplementary Fig. 1b–f)[1]. The conservation of this architecture throughout the DASS/SLC13 family has been confirmed by X-ray crystallography and cryo-electron microscopy (cryo-EM) structures of VcINDY, LaINDY, a dicarboxylate exchanger from *Lactobacillus acidophilus*, and the human citrate transporter NaCT[1,4,19,20]. Comparison of VcINDY in its inward-facing ($C_i$) conformation with the outward-facing ($C_o$) structure of LaINDY, along with MD simulations, reveals that an elevator-type movement of the transport domain, through an ~12 Å translation along with an ~35° rotation, facilitates translocation of the substrate from one side of the membrane to the other[19]. In fact, the structural and mechanistic conservation may extend beyond DASS to the broader Ion Transport Superfamily (ITS)[5,21].

Substrate transport of VcINDY is driven by the inwardly-directed Na$^+$ gradient, with dicarboxylate import coupled to the co-transport of three sodium ions (Supplementary Fig. 1a, b)[1,2,22]. The binding sites for the substrate and two central Na$^+$s have been identified in the structures of VcINDY in its Na$^+$- and substrate-bound inward-facing ($C_i$-Na$^+$-S) state (Supplementary Fig. 1e, f)[1,4]. The Na1 site on the N-terminal half of the transport domain is defined by a clamshell formed by loop L5ab and the tip of hairpin HP$_{in}$. A second clamshell encloses Na2, related to Na1 by inverted-repeat pseudo-symmetry in the sequence and structure, and formed by L10ab and the tip of hairpin HP$_{out}$ (Supplementary Fig. 1c). In addition to binding the Na$^+$s, both hairpin tips also form parts of the substrate-binding site, located between the Na$^+$ sites. Each hairpin tip consists of a conserved Ser-Asn-Thr (SNT) motif, and the two SNT motifs form part of the substrate-binding site, making direct contact with carboxylate groups of the substrate. Whereas these two SNT signature motifs are responsible for recognizing carboxylate, additional residues in neighboring loops have been proposed to distinguish between different kinds of substrates[4].

Furthermore, VcINDY's structure with sodium in the absence of a substrate (the $C_i$-Na$^+$ state), determined in 100 mM Na$^+$, is very similar to that of the Na$^+$- and substrate-bound state $C_i$-Na$^+$-S[19].

While the Na$^+$- and substrate-binding sites in VcINDY have been well-characterized[1,4,23], the coupling mechanism between the electrochemical gradient and substrate transport[24] is less well understood. There is strong evidence that charge compensation by sodium ions is essential in lowering the energy barrier for transporting the di- and trivalent anionic substrates across the membrane[19]. However, such charge compensation alone does not necessarily result in substrate binding as Li$^+$ is able to bind to VcINDY similarly to Na$^+$, but results in a lower affinity substrate binding site and considerably reduced transport rates[2,23]. More importantly, charge neutralization cannot explain the sequential binding observed for VcINDY. As is the case for other DASS proteins[25–28], all available experimental evidence from both whole cells and reconstituted systems supports the notion that in VcINDY sodium ions and substrate bind in a sequential manner, with Na$^+$s binding first, followed by dicarboxylate[2,3,23,29]. As a secondary-active transporter can transport substrate in either direction, it follows that the release of the substrate and Na$^+$s is also ordered, with the substrates being released first.

Structures of VcINDY in the Na$^+$- and substrate-bound state $C_i$-Na$^+$-S, in which the Na$^+$ sites share residues with the substrate site in their center, allowed us to propose that substrate binding in VcINDY follows a cooperative binding mechanism via conformational selection[1,30]. In this mechanism, the binding of sodium ions helps to induce a proper binding site for the substrate (Supplementary Fig. 1a, b). Conversely, in the absence of bound sodium ions the substrate-binding site will change significantly, such that the substrate cannot bind. Not only can such a mechanism be part of Na$^+$—substrate coupling, it may also explain the sequential binding observed for VcINDY.

This conformational selection mechanism of substrate binding enables us to make two explicit, experimentally testable predictions. First, the affinity of the transporter to a substrate must be much higher in the presence of Na$^+$ than in its absence. Second, substantial structural changes will occur at the Na$^+$ sites in the absence of sodium, affecting substrate binding.

In this work, we aim to test these two predictions using a combination of structure determination by single-particle cryo-EM, substrate-binding affinity measurements by intrinsic tryptophan fluorescence quenching, and position accessibility quantification via a newly-developed site-directed cysteine alkylation assay[29]. In particular, we characterize VcINDY in sodium-saturating and sodium-free conditions, including structures in $C_i$-Na$^+$ and $C_i$-*apo* states. These experimental results allow us to directly test the conformational selection binding model of VcINDY.

## Results

**Succinate binding depends on the presence of Na$^+$.** To test the first of the predictions generated from our conformational selection hypothesis, we measured VcINDY's binding affinity for the model substrate, succinate, in both the presence and absence of Na$^+$ (Supplementary Fig. 2). We reasoned that VcINDY's tryptophans, particularly Trp148 located at the tip of HP$_{in}$ of the Na1 site, may change its position or environment upon Na$^+$-/substrate-binding. Thus, we used intrinsic tryptophan fluorescence quenching, a technique that has been successfully applied to measure substrate binding for various membrane transporters[20,31–36]. In the presence of 100 mM Na$^+$, detergent-purified VcINDY was found to bind succinate with an apparent $K_d$ of 92.2 ± 47.4 μM (Fig. 1a, Supplementary Fig. 2b). For comparison, the human NaCT in the same protein family binds its substrate citrate at an apparent $K_d$ of 148 ± 28 μM[20].

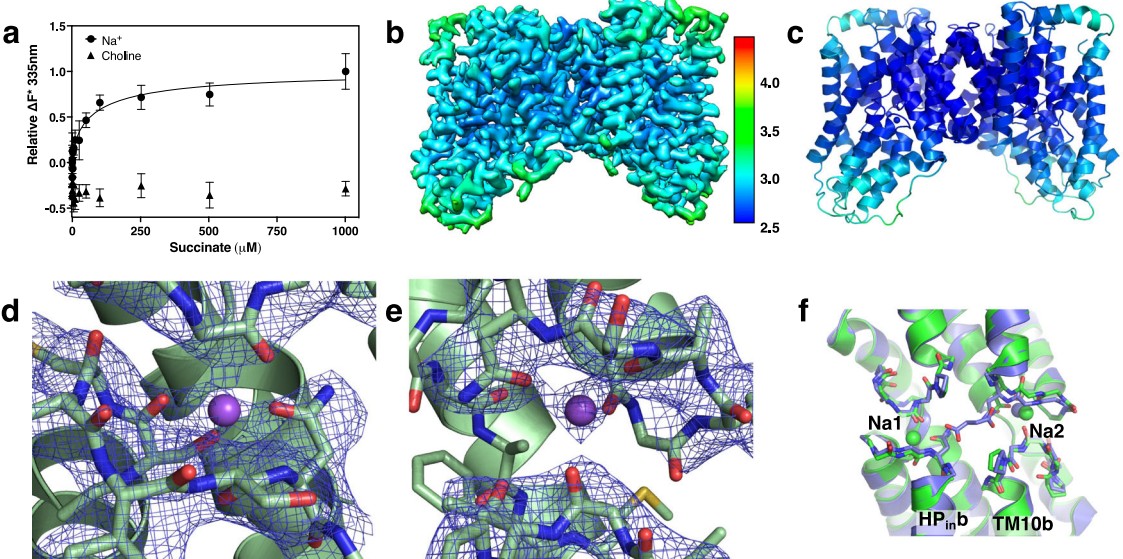

**Fig. 1 Cryo-EM structure of VcINDY in the $C_i$-Na$^+$ state determined in 300 mM Na$^+$. a** Measurements of succinate binding to detergent-purified VcINDY in the presence of 100 mM NaCl, using intrinsic tryptophan fluorescence quenching ($N = 4$). Data are presented as mean values ± SEM. The apparent $K_d$ was determined to be 92.2 ± 47.4 mM. When NaCl was replaced with Choline chloride, no binding of succinate to VcINDY could be measured ($N = 4$). **b** Cryo-EM map of VcINDY determined in the presence of 300 mM NaCl. The map is colored by local resolution (Å) and contoured at 5.1 σ. The overall map resolution is 2.83 Å. **c** Structure of VcINDY in the $C_i$-Na$^+$ state. The structure is colored by the B-factor. **d** Na1 site structure and Coulomb map. **e** Na2 site structure and Coulomb map. **f** Overlay of VcINDY structures around the substrate and sodium binding sites in the $C_i$-Na$^+$ state (green) and $C_i$-Na$^+$-S state (PDB ID: 5UL7, blue). There is very little structural change observed between the two states.

To measure the binding affinity of succinate to VcINDY with empty Na$^+$ binding sites, we searched for a cation to replace Na$^+$ in the purification buffer. This ion should not occupy the Na1 or Na2 sites while still allowing the transporter protein to remain stable in the solution. K$^+$ is unable to power substrate transport in VcINDY, but was found to be unsuitable as the protein precipitated when purified in the presence of 100 mM KCl. We next tested the organic cation choline ($C_5H_{14}NO^+$, Ch$^+$ in abbreviation). We reasoned that Ch$^+$ would be more stabilizing than K$^+$ based on its position in the Hofmeister series[37], and that its size would preclude it from occupying Na$^+$ binding sites[38]. Indeed, VcINDY purified in 100 mM NaCl remained soluble at 0.5–1.0 mg/mL after diluting the sample 11,000-fold in 100 mM ChCl. VcINDY was therefore purified in the presence of 100 mM Ch$^+$ as the only monovalent cation. The protein eluted as a sharp, symmetrical peak on a size-exclusion chromatography column (Supplementary Fig. 2a), confirming its stability and structural homogeneity.

Notably, intrinsic tryptophan fluorescence quenching with VcINDY purified and assayed in the presence of 100 mM Ch$^+$ revealed no succinate binding (Fig. 1a, Supplementary Fig. 2c). Thus, the binding measurements in the presence and absence of Na$^+$ are consistent with a conformational selection model where bound sodium ions are necessary to VcINDY forming a proper binding site for succinate. Encouraged by these findings and our ability to produce stable, structurally homogeneous and Na$^+$-free VcINDY, we next sought to uncover the structural basis of this Na$^+$—substrate coupling by determining the transporter's structures using cryo-EM in different states.

**Structure of VcINDY in 300 mM Na$^+$.** Generally speaking, the transport mechanism of a secondary-active transporter is reversible, in which the direction of substrate translocation depends on the direction and magnitude of the driving force (Supplementary Fig. 1a, b). Consequently, substrate binding is structurally equivalent to substrate release. Therefore, to provide structural insights into VcINDY's binding process, we aimed to characterize

the substrate release process in the inward-facing ($C_i$) conformations by capturing the structures of VcINDY in the following states: its Na$^+$-and substrate-bound state ($C_i$-Na$^+$-S), its Na$^+$-bound state ($C_i$-Na$^+$) and its Na$^+$- and substrate-free state ($C_i$-apo).

The $C_i$-Na$^+$-S structure of VcINDY has previously been solved using X-ray crystallography[1,4]. Additionally, we had characterized the $C_i$-Na$^+$ state using a cryo-EM structure of VcINDY purified in 100 mM Na$^+$ without substrate[19]. However, as the apparent $K_{0.5}$ for Na$^+$ for VcINDY was measured to be 41.7 mM[2], our earlier VcINDY sample in 100 mM Na$^+$ likely represents a mixture of the $C_i$-Na$^+$ and $C_i$-apo states. It is unclear whether the subsequent cryo-EM image processing of the particles was able to exclude all particles of the Na$^+$-free $C_i$-apo state. To more clearly and definitively resolve the $C_i$-Na$^+$ state structure, in the current work we purified and determined a structure of VcINDY in 300 mM Na$^+$. This ion concentration was optimized to increase the Na$^+$ occupancy, while, at the same time, ensuring a low enough noise level in the cryo-EM images to determine a $C_i$-Na$^+$ state structure of this small membrane protein (total dimer mass: 126 kDa) at 2.83 Å resolution (Fig. 1b, c, Supplementary Figs. 3 and 4a–c, Table 1).

Compared with the two previously determined cryo-EM structures of VcINDY in the presence of 100 mM NaCl[19], the herein reported structure in 300 mM Na$^+$ (Fig. 1c, Supplementary Fig. 4b) is identical to the one bound to a Fab and embedded in lipid nanodisc (PDB ID: 6WW5)[19] (r.m.s.d. of 0.460 Å for all the non-hydrogen atoms), except for the position of the last three residues at the C-terminus, which interact with the Fab molecule used for structure determination (Supplementary Fig. 4d). Furthermore, though the map obtained in 300 mM Na$^+$ conditions clarified the loop connecting HP$_{out}$b and TM10b, the model in 300 mM NaCl is effectively identical to the other previous $C_i$-Na$^+$ structure in 100 mM NaCl, determined in amphipol and without Fab (PDB ID: 6WU3)[19], with an r.m.s.d of 0.358 Å after excluding Val392 – Pro400 (Supplementary Fig. 4d).

As expected from the higher Na$^+$ occupancy in the 300 mM sample, better-defined densities appeared within both the Na1 and Na2 clamshells (Fig. 1d, e), which were absent in the previous 100 mM Na$^+$ maps[19]. In addition to coordination by side chains and backbone carbonyl oxygens, the sodium ion at the Na1 site is stabilized by the helix dipole moments from HP$_{in}$b and TM5b (Fig. 1f; Supplementary Fig. 1e, f), as previously observed in other membrane proteins[39,40]. Similarly, the Na$^+$ ion in the Na2 site is stabilized by HP$_{out}$b and TM10b.

Finally, this higher resolution map confirmed our earlier observations that succinate release caused only limited changes at the substrate-binding site without relaxing the two Na$^+$ clamshells[19]. Both the overall structure and the sodium- and substrate-binding sites in the C$_i$-Na$^+$ state are similar to those in the sodium- and substrate-bound C$_i$-Na$^+$-S state (Fig. 1e, f, Supplementary Fig. 4e). The similarity of these structures agrees with our conformational selection model of Na$^+$ – substrate coupling, which requires sodium-binding induce a C$_i$-Na$^+$ state structure that can bind substrate directly as in the C$_i$-Na$^+$-S state (Supplementary Fig. 1a, b).

**Apo structure of VcINDY in Choline$^+$.** With the structures of sodium- and succinate-bound[1,4] and Na$^+$-only bound (Fig. 1) states in hand, the missing piece of the puzzle to validate the Na$^+$ conformational selection mechanism was the C$_i$-apo state structure of the transporter protein. As a Ch$^+$ ion is too large to fit into a Na$^+$ binding site[36,38], and VcINDY was stable and monodisperse in the presence of 100 mM ChCl (Supplementary Fig. 2a), such a preparation allowed us to obtain cryo-EM maps of the C$_i$-apo state (Fig. 2, Supplementary Figs. 5 and 6, Table 1). Unlike the VcINDY map in 300 mM Na$^+$ for which 3D classification converged to a single map (Supplementary Fig. 3), the VcINDY-choline dataset yielded four distinct classes at a resolution range of 3.6—4.4 Å resolution (Supplementary Figs. 5 and 7). The 3D class with the highest resolution was further refined to 3.23 Å resolution (Fig. 2a, b, Supplementary Fig. 6c). The least well-resolved regions of the map, and highest B-factors of the model, are found in L4-HP$_{in}$ and L9-HP$_{out}$, two previously-identified hinge regions that facilitate movement of the transport domain[19]. Whereas the overall fold of the protein in Ch$^+$ remains the same (Supplementary Fig. 6d, f), Na$^+$ densities within the Na1 and Na2 clamshells are totally absent. Additional local changes are observed for the protein parts near the Na1 and Na2 sites (Fig. 2c), with a loss of density in each C$_i$-apo map at the HP$_{in}$b and TM10b helices (Fig. 3b), indicating increased local structural flexibility.

**Flexibility of Apo VcINDY near the Na1 and Na2 sites.** While the VcINDY C$_i$-apo state overall structure is similar to those in

the 300 mM Na$^+$ (r.m.s.d of 0.672 Å) (Supplementary Fig. 6f), the model exhibited significant changes near the Na1 and N2 sites (Figs. 2c and 3a, b). The tip of HP$_{in}$ and the L10a-b loop have moved away from the Na1 and Na2 sites, respectively, with the carbonyls of Ala376 and Ala420, and the side chain of Asn378 also rotated away from the Na2 site. Notably, HP$_{in}$b near the Na1 site and TM10b near the Na2 site and their connecting loops showed marked decreased density in the cryo-EM map, corresponding to the increased flexibility of these regions (Fig. 3a, b). Correspondingly, the model exhibited significantly higher relative B-factors in the same regions compared to the rest of the model (Fig. 3c, d). However, we recognized that such a single, averaged model might not fully describe the true structural ensemble, and sought a method to describe the C$_i$-apo state's mobility.

To further analyze such local flexibility, we used simulated annealing[41,42] in a model refinement protocol analogous to protein structure determination by NMR spectroscopy[43]. We reasoned that in multiple, separate refinements with simulated annealing the rigid parts of the VcINDY would converge to the same coordinates, while mobile portions of the protein would arrive at distinct atomic positions in each run. We term this as NMR-style analysis in recognition of NMR's power to characterize protein dynamics, though in cryo-EM the constraints are Coulomb potential maps rather than distances.

Most parts of the VcINDY structure exhibit no variation in the C$_i$-Na$^+$ state, including HP$_{in}$b and TM10b in the 300 mM Na$^+$ condition (Fig. 3e, Supplementary Fig. 8a). In contrast, in the C$_i$-apo state the NMR-style analysis clearly illustrated the structural heterogeneity near the Na1 and Na2 sites (Fig. 3f, Supplementary Fig. 8b). Instead of converging to one structure, the simulated annealing resulted in an ensemble of structures, with the greatest variations occurring in the HP$_{in}$b and TM10b regions. The mean r.m.s.d. of the transport domain's backbone atoms for the C$_i$-apo protomers is 0.589 Å, as opposed to 0.099 Å among C$_i$-Na$^+$ protomers refined using the same protocol to the same resolution. As the 3.23 Å apo map imposes C2 symmetry on one of four classes of particles in ChCl (Supplementary Fig. 5), and all four classes are different (Supplementary Fig. 7), the degree of flexibility of these helices in the C$_i$-apo state is likely to be even greater. Such helix flexibility results from the absence of Na$^+$ interactions with residues in the clamshells and with the dipoles of HP$_{in}$b and TM10b[44,45].

**Site-directed alkylation supports structural changes to Na1 and Na2 sites.** To confirm the local conformational changes and helix flexibility observed in our VcINDY structures, we implemented a site-directed cysteine alkylation strategy that can directly assess the solvent accessibility of specific positions in a protein. In this

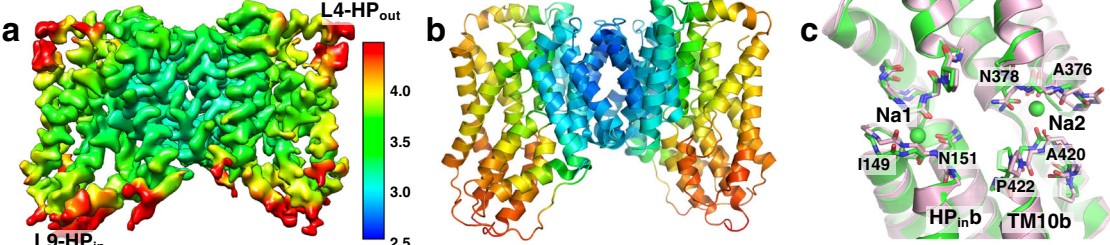

**Fig. 2 Cryo-EM structure of VcINDY in the C$_i$-apo state determined in Choline$^+$. a** Cryo-EM map of VcINDY preserved in amphipol determined in the presence of 100 mM Choline Chloride. The map is colored by local resolution (Å) on the same scale as Fig. 1b and contoured at 4.8 σ. The overall map resolution is 3.23 Å. The two previously-identified hinge regions which facilitate movement of the transport domain[19], L4-HP$_{in}$ and L9-HP$_{out}$, are found to be most flexible. **b** Structure of VcINDY in the C$_i$-apo state. The structure is colored by the B-factor on the same scale as Fig. 1b. **c** Overlay of VcINDY structures around substrate and sodium binding sites in the sodium-bound C$_i$-Na$^+$ state (green) and the C$_i$-apo state (pink). The structures of the two Na1 and Na2 clamshells have changed in the absence of sodium ions, particularly around residues Ile149, Asn151, Ala376, Asn378, Ala420 and Pro422.

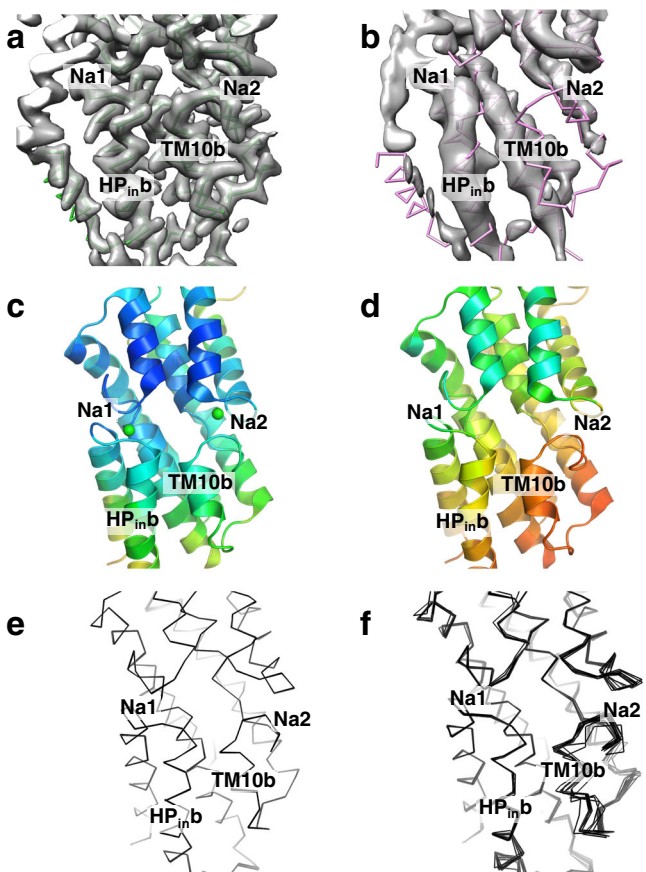

**Fig. 3 VcINDY flexibility changes near the Na1 and Na2 sites between the $C_i$-Na$^+$ state and $C_i$-apo states. a** Cryo-EM density map in 300 mM NaCl. **b**. Cryo-EM density map in 100 mM Choline Chloride. In **a** and **b**, the respective protein models' backbones are fitted into the densities. Maps are contoured such that the scaffold domains have equal volume. **c**. Structure of VcINDY in its $C_i$-Na$^+$ state. **d**. Structure of VcINDY in its $C_i$-apo state. In **c** and **d**, the structures are colored by normalized B-factors. **e** NMR-style analysis of the VcINDY structure in Na$^+$. **f** NMR-style analysis of the VcINDY structure in Choline$^+$. The resolution for refinement of both structures in **e** and **f** was truncated to 3.23 Å. In the absence of sodium, the helices on the cytosolic side of Na1 and Na2, particularly HP$_{in}$b and TM10b and their connecting loops, show markedly increase flexibility. Instead of a single structure, the $C_i$-apo model consists of an ensemble of structures.

approach, single cysteines are introduced into a Cys-less version of VcINDY, which is capable of robust Na$^+$-driven transport[2,3]. Following purification, the cysteine mutants of VcINDY are incubated with the thiol-reactive methoxypolyethylene glycol maleimide 5 K (mPEG5K). This tag reacts with solvent-accessible cysteines and increases the protein mass by ~5 kDa, which is separable from unmodified protein on an SDS-PAGE gel. As mPEG5K will react faster with cysteines that are more accessible, monitoring PEGylation over time provides us with the ability to follow changes in the accessibility of particular parts of the protein under different conditions[29].

To test our conformational selection model using biochemical approaches, we designed a panel of single-cysteine mutants of VcINDY that would report on the Na$^+$-dependent accessibility changes at the Na1 and Na2 sites predicted from structures (Fig. 4a, Supplementary Fig. 8d). We selected residues that, if our cooperative binding model is accurate, will exhibit a higher rate of PEGylation in the absence of Na$^+$ compared to its presence due to the increased mobility of HP$_{in}$ and TM10b. To create our panel proximal to the Na1 site, we purified four cysteine mutants whose

reactive thiol groups are buried in the $C_i$-Na$^+$ state behind HP$_{in}$ (L138C on HP$_{in}$a, A155C and V162C on HP$_{in}$b and A189C on TM5a). However, similar cysteine substitutions near Na2 (Val427, Ile433, Gly442 and Met438) resulted in diminished expression levels, likely indicating the importance of these residues to the stability of the protein. Fortunately, cysteine mutation of Val441 to cysteine, a residue located on TM11 and behind TM10b (Fig. 4a, Supplementary Fig. 8d), expressed well and allowed for purification. Typically, well-expressing single cysteine VcINDY mutants that can be purified are capable of Na$^+$-driven succinate transport[29].

We monitored the PEGylation of each mutant in the presence and absence of Na$^+$. Under these reaction conditions there is no PEGylation of the Cys-less variant, demonstrating no background labelling that could hinder analysis (Fig. 4b, top row). In the presence of 300 mM Na$^+$ we observed complete inhibition of PEGylation at every position (Leu138, Ala155, Val162, Ala189 and Val411) over the time course of 60 min (Fig. 4b, left panels), in agreement with our model that these residues are buried in the Na$^+$-bound state. However, in the absence of Na$^+$ (but with 300 mM Ch$^+$), every mutant showed escalated levels of PEGylation over time (Fig. 4b, right panels), indicating the increased flexibility of HP$_{in}$b and TM10b.

To ensure that the change in PEGylation rate that we observed was due to changes in residue accessibility and not caused by an unforeseen effect the cations may have on the PEGylation reaction, we monitored the reaction rate of a position for which we observed no accessibility change in the structural analysis. A cysteine mutant at Ser436, positioned at the periphery of the transporter protein (Fig. 4a), exhibited minimal Na$^+$-dependent accessibility changes (Fig. 4b, bottom row).

These accessibility measurements, along with our previous PEGylation results on three other VcINDY residues near the Na1 site (T154C, M157C and T177C, Supplementary Fig. 8e)[29], fully support the changes in protein dynamics predicted upon occupation of the Na1 and Na2 sites, and are consistent with a conformational selection coupling model.

**Structural comparison of $C_i$-Na$^+$-S, $C_i$-Na$^+$ and $C_i$-apo states.** The VcINDY structures determined in 300 mM Na$^+$ and apo as reported here, together with previously-determined X-ray structure of the protein with both sodium and substrate bound[1,4], allowed us to examine the structural changes of the transporter between the $C_i$-Na$^+$-S, $C_i$-Na$^+$ and $C_i$-apo states. In addition to the flexibility observed in HP$_{in}$b and TM10b, we observed amino acid sidechain movements both at the interface between the scaffold domain and the transport domain, as well as on the periplasmic surface of the protein.

At the scaffold–transport domain interface, side chains of several bulky amino acids rotated or shifted between the three states, including Phe100, His111 and Phe326 (Fig. 5a). On the periplasmic surface, Trp461 at the C-terminus is buried in the apo and Na$^+$-bound structures (Fig. 5b). However, in the $C_i$-Na$^+$-S structure, the ring of the nearby Phe220 was rotated by ~90°, pushing out the side chain of Trp461, leaving the C-terminus pointing to the periplasmic space. Accordingly, the loop between HP$_{out}$b and TM10a moved into the periplasmic space of the apo VcINDY structure, displacing Glu394 and breaking its salt bridge with Lys337. Whereas no single switch was identified that can trigger conformational exchanges between the inward- and outward-facing states, local structural changes observed here suggest that small changes at multiple locations are required for inter-conformation transitions in VcINDY.

In comparing maps of the three states, we noted the VcINDY $C_i$-Na$^+$ map reported herein was sufficiently detailed to identify

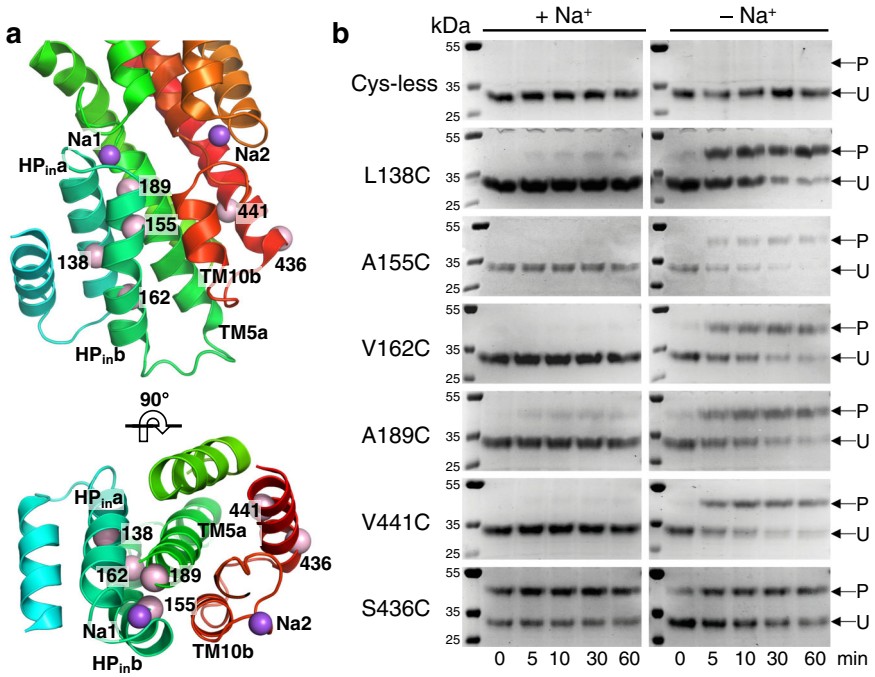

**Fig. 4 Cysteine alkylation with mPEG5K of VcINDY near the Na1 and Na2 sites in the presence and absence of Na⁺. a** Location of cysteine mutations. Our structures suggested that $HP_{in}b$ and TM10b become flexible in the absence of sodium, increasing the solvent accessibility of Leu138, Ala155, Val162 and Ala189 near the Na1 site, and Val441 near the Na2 site. Position Ser436, for which no accessibility change was observed between our structures, is used as a control. On a Cys-less background, residues at these positions were individually mutated to a cysteine for mPEG5K labeling. For clarity, only amino acid numbers are labeled and the types are omitted. **b** Coomassie Brilliant Blue-stained non-reducing polyacrylamide gels showing the site-directed PEGylation of each cysteine mutant over time in the presence and absence of Na⁺. P: PEGylated protein; U: Un-PEGylated protein. Each reaction was performed on two separate occasions with the same result. Source data is provided as Source Data file.

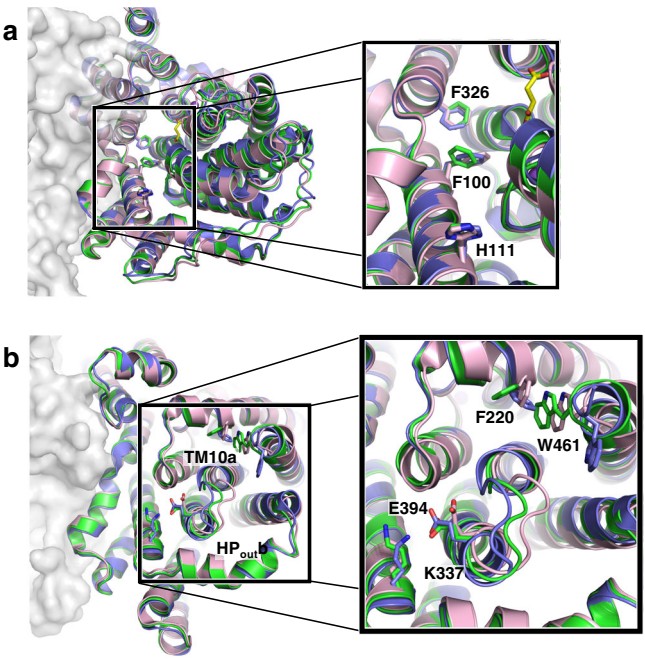

**Fig. 5 Movement of VcINDY's amino acid side chains between its $C_i$-Na⁺-S, $C_i$-Na⁺ and $C_i$-apo states.** VcINDY structures in three states are overlaid: $C_i$-Na⁺-S (blue), $C_i$-Na⁺ (green) and $C_i$-apo (pink) states. **a** At the scaffold-transport domain interface, the side chains of Phe100, His111 and Phe326 rotate between states. **b** On the periplasmic surface, some loops and side chains move between the states, including Phe220, Lys337, Glu394 and Trp462.

five ordered water molecules buried at the dimer interface (Supplementary Fig. 8c). The water molecules are not visible in previous maps, or the VcINDY-apo map, indicating the high-resolution of the VcINDY $C_i$-Na⁺ map reported here was necessary for their identification. These waters are arranged in a square pyramidal configuration in the largely hydrophobic pocket, coordinated by only the symmetry-related carbonyls of Phe92 and inter-water hydrogen bonds. The role of these waters in VcINDY folding or transport are unclear, though protein folding defects underlie several pathogenic mutations on the equivalent dimerization interface of NaCT[5].

## Discussion

Despite great advances in structural and mechanistic studies on membrane transporters over the past twenty years[46–50], the ion–substrate coupling mechanism is well characterized for only very few co-transporters, limiting our understanding of this fundamental aspect of the secondary-active transport mechanism. Here, we have described the structural basis of ion–substrate coupling for VcINDY, which reveals a distinct conformational selection mechanism that ensures obligatory coupling.

While Na⁺ sites in some other Na⁺-dependent transporters are buried in the middle of the protein[38,48,49,51], the sodium sites in VcINDY are directly accessible from the extramembrane space. Previous experimental data support that Na⁺-driven DASS co-transporters operate via an ordered binding and release[2,25–29]. Specifically, Na⁺ binding occurs before substrate binding, while substrate release precedes Na⁺ release. For VcINDY, we have now observed that sodium release from the Na1 and Na2 sites in the cytoplasm allows increased conformational diversity going from the $C_i$-Na⁺ to the $C_i$-apo states, whereas the $C_i$-Na⁺ and $C_i$-

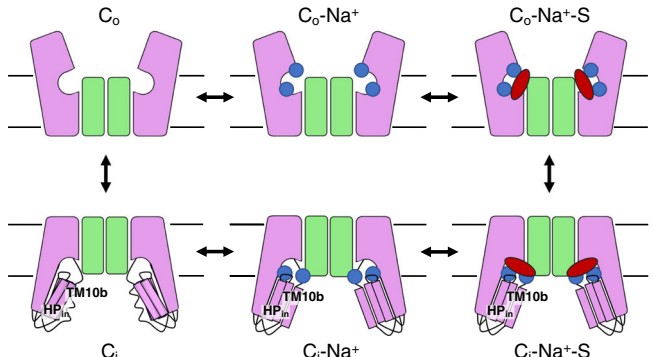

**Fig. 6 Schematic model of conformational selection mechanism for sodium—substrate coupling in VcINDY.** In the absence of sodium ions, $HP_{in}b$ and TM10b, along with their connecting loops responsible for sodium and substrate binding, are flexible. From the ensemble of flexible structures, the binding of sodium ions (blue circles) selects a conformation with a proper binding site for the substrate, allowing its binding (red oval). The scaffold and transport domains in each protomer are colored as green and pink, respectively. Only the Na1 and Na2 sites are illustrated. Transport domain movements in the two protomers are shown as symmetric for simplicity but are functionally independent.

$Na^+$-S states are structurally similar (Fig. 6). Specifically, the movement of helices $HP_{in}b$ and TM10b is tightly coupled to $Na^+$ binding. At the Na1 and Na2 sites, the sodium ions are stabilized via direct and ion—dipole interaction with the two helices. Therefore, upon $Na^+$-release, the elimination of these interactions caused the relaxation of the $HP_{in}b$ and TM10b helices[44,45], leading to increased mobility in the connected loops responsible for substrate binding. In the reverse reaction, ions binding at the Na1 and Na2 sites, concurrent with helix re-ordering, select from the ensemble a structure with the proper binding site for the substrate. While the effects of VcINDY's cryptic third $Na^+$ are still to be determined, we now have established a structural understanding of the $Na^+$—substrate coupling mechanism for this co-transporter. By extension, other DASS transporters may utilise a similar structural mechanism for $Na^+$—substrate coupling (Fig. 6).

The structural basis of $Na^+$-substrate coupling for VcINDY is distinct from that of $Glt_{Ph}/Glt_{Tk}$ from the dicarboxylate/amino acid:cation symporter family which otherwise share several commonalities with VcINDY including the presence of re-entrant hairpin loops and the utilization of an elevator-like mechanism[1,3,4,48,52–56]. In addition, as we have shown here for VcINDY, a cooperative binding mechanism has been suggested for both $Glt_{Ph}$ and $Glt_{Tk}$, which requires the initial binding of $Na^+$ in order to prime the binding site for the substrate, aspartate[38,57]. However, the structural basis of $Na^+$-substrate coupling in $Glt_{Ph}/Glt_{Tk}$ differs substantially from the coupling mechanism we observe for VcINDY. Rather than the general relaxation of a helix governing substrate-binding site formation, the binding of $Na^+$ to $Glt_{Ph}/Glt_{Tk}$ induces discrete conformational changes of a small number of amino acid residues centered on the highly conserved NMDGT motif[53,57,58]. As is the case here for VcINDY (Fig. 6), the fully loaded and $Na^+$-only bound structures of $Glt_{Ph}/Glt_{Tk}$ are largely identical[52,53,57,58], demonstrating that $Na^+$ binding drives the formation of the substrate-binding site, and not the substrate itself.

In addition to conformational selection, another mechanism for ion–substrate coupling of co-transporters has been proposed to be charge compensation[5,19]. Such a mechanism can greatly minimize the energy penalty for translocating charged substrates across the hydrophobic lipid bilayer[59,60]. Unlike for DASS exchangers[19], where charge compensation is the major force for overcoming the energy barrier in the $C_o \leftrightarrow C_i$ transition, both local structural ordering and charge balance are needed for $Na^+$-coupled co-transporters within the DASS family.

Comparison of the VcINDY structures reported here with those determined earlier[1,4,19], of three states in total, also sheds new light on the mechanism of the transporter's conformational switching between the two sides of the membrane. As the $C_i$-*apo* structure is significantly different from that of the $C_i$-$Na^+$-S state, their corresponding transitions to the outward-facing state: $C_i$-*apo* to $C_o$-*apo* and $C_i$-$Na^+$-S to $C_o$-$Na^+$-S, are different at the transport-scaffold domain interface (Fig. 6). Whereas the transition between $C_i$-$Na^+$-S and $C_o$-$Na^+$-S state can be described as rigid-body movement, as was seen in the DASS exchangers[5], the co-transporters' $C_o$-*apo* $\leftrightarrow$ $C_i$-*apo* state transition likely involves large structural rearrangements of the transport domain. Considering the pseudo-symmetry of the DASS fold, the $C_i$-*apo* $\rightarrow$ $C_o$-*apo* movement would require refolding of TM10b to pack against the scaffold domain, and possibly the concurrent unfolding of TM5b. This potential asymmetry between the *apo*-state transition ($C_o$-*apo* $\leftrightarrow$ $C_i$-*apo*) and transition of the fully-loaded transporter ($C_i$-$Na^+$-S $\leftrightarrow$ $C_o$-$Na^+$-S) needs further investigation. Finally, the pseudo-symmetry within the DASS fold and sequence[1,3,19] and $Na^+$ dependent solvent accessibility of the S381C mutant on $HP_{out}b$ of VcINDY, which we investigated previously[29], seem to indicate the $C_o$ state also undergoes $Na^+$ dependent conformational selection to enable substrate binding. However, verifying this hypothesis will require structural characterization of a DASS symporter's outward-facing state.

## Methods

**VcINDY expression and purification.** Expression and purification of VcINDY were carried out according to our previous protocol[1]. Briefly, *E. coli* BL21-AI cells (Invitrogen) were transformed with a modified pET vector[61] encoding N-terminal 10x His tagged VcINDY. Cells were grown at 32 °C until $OD_{595}$ reached 0.8, protein expression occurred at 19 °C following IPTG induction, and cells were harvested 16 h post-induction. Cell membranes were solubilized in 1.2 % DDM and the protein was purified on a $Ni^{2+}$-NTA column. For cryo-EM and substrate binding experiments, the protein was purified using size-exclusion chromatography (SEC) in different buffers. The protein used for the cysteine alkylation assays was produced as described previously[29].

**Tryptophan fluorescence quenching assay.** Tryptophan fluorescence quenching was used to measure affinity of succinate to purified VcINDY in detergent, using a protocol adapted from earlier work on other membrane transporters[20,31–34]. VcINDY purified by SEC in a buffer of 25 mM Tris pH 7.5, 100 mM NaCl and 0.05% DDM was used to measure succinate affinity, while the 100 mM NaCl was replaced by 100 mM ChCl for affinity measurements in the absence of sodium. Protein was diluted to a final concentration of 4 μM in SEC buffer. Using a Horiba FluoroMax-4 fluorometer (Kyoto, Japan) at 22 °C and a 280 nm excitation wavelength, the emission spectrum was recorded between 290 and 400 nm. The emission maximum was determined to be 335 nm. Subsequently, the change in fluorescent emission at 335 nm was monitored with increasing concentrations of succinic acid (pH 7.5), from 0.1 μM to 1 mM. Each experimental condition was repeated 4 times. The binding curve was fit in Prism using a quadratic binding equation to account for bound substrate[62].

**Amphipol exchange and cryo-EM sample preparation.** From $Ni^{2+}$-NTA purified VcINDY, DDM detergent was exchanged to PMAL-C8 (Anatrace, Maumee, OH) as previously described[19,63]. Following further purification by SEC in buffer containing 25 mM Tris pH 7.5, 100 mM NaCl and 0.2 mM TCEP, the NaCl concentration was increased to 300 mM and the protein sample was concentrated to 1.3 mg/mL. For the *apo* protein preparation, NaCl in the abovementioned SEC buffer was replaced with 100 mM ChCl, and the protein sample was concentrated to 1.3 mg/mL.

Cryo-EM grids were prepared by applying 3 μL of protein to a glow-discharged QuantiAuFoil R1.2/1.3 300-mesh grid (Quantifoil, Germany) and blotted for 2.5 to 4 s under 100% humidity at 4 °C before plunging into liquid ethane using a Mark IV Vitrobot (FEI).

**Cryo-EM data collection.** Cryo-EM data were acquired on a Titan Krios microscope with a K3 direct electron detector, using a GIF-Quantum energy filter with a

20-eV slit width. SerialEM was used for automated data collection[64]. Each micrograph was dose-fractioned over 60 frames, with an accumulated dose of 65 e⁻/Å². 

**Cryo-EM image processing and model building**. Motion correction, CTF estimation, particle picking, 2D classification, ab initio model generation, heterogenous and non-uniform refinement, and per-particle CTF refinement were all performed with cryoSPARC[65]. Each dataset was processed using the same protocol, except as noted.

Micrographs underwent patch motion correction and patch CTF estimation, and those with an overall resolution worse than 8 Å were excluded from subsequent steps. An ellipse-based particle picker identified particles used to generate initial 2D classes. These classes were used for template-based particle picking. Template identified particles were extracted and subjected to 2D classification. A subset of well-resolved 2D classes were used for the initial ab initio model building, while all picked particles were subsequently used for heterogeneous 3D refinement. After multiple rounds of 3D classification (ab initio model generation and heterogeneous 3D refinement with two or more classes), a single class was selected for nonuniform 3D refinement with C2 symmetry imposed, resulting in the final map.

All Cryo-EM maps were sharpened using Auto-sharpen Map in Phenix[66], models were built in Coot[67], and refined in Phenix real space refine[68]. The model for VcINDY in NaCl was built using the structure of VcINDY embedded in a lipid nanodisc (PDB: 6WW5) as an initial model, with lipid and antibody fragments removed. The VcINDY model in choline used the structure of VcINDY in 300 mM NaCl, with ions and waters removed, as the starting model.

The NMR-style analysis used 5 independent runs of phenix.real_space_refine[66] to refine the models of VcINDY in *apo* and in 300 mM NaCl, with ions and waters removed, using unique computational seeds for each run. Each refinement was performed with simulated annealing, without NCS constraints or secondary structure restraints, and a refinement resolution limit of 3.23 Å for both maps. Analysis with or without map sharpening, or randomizing initial atomic positions using phenix.pdbtools, gave similar results. Transport domain maps were scaled to equivalent contours using the scaffold domain's volume as an internal standard after extracting with phenix.map_box. Figures were made using UCSF Chimera[69] and PyMOL[70].

**Cysteine alkylation assay**. For the cysteine alkylation experiments, each purified cysteine mutant was exchanged into reaction buffer containing 50 mM Tris, pH 7, 5% glycerol, 0.1% DDM and either 300 mM NaCl or 300 mM ChCl (Na⁺-free conditions). Protein samples were incubated with 6 mM mPEG5K and samples were taken at the indicated timepoints and immediately quenched by addition of SDS-PAGE samples buffer containing 100 mM methyl methanesulfonate (MMTS). Samples were analyzed with Coomassie Brilliant Blue-stained non-reducing polyacrylamide gels.

**Reporting summary**. Further information on research design is available in the Nature Research Reporting Summary linked to this article.

## Data availability

The cryo-EM particle stacks, maps and models generated in this study have been deposited in EMPIAR image archive, EMDB database and the Protein Data Bank, respectively, under accession codes EMPIAR-10969, EMD-25757 and PDB-7T9G) for VcINDY-Na⁺ (300 mM) structure and under accession codes EMPIAR-10970, EMD-25756 and PDB-7T9F) for VcINDY-Ch⁺ structure. Source Data for Fig. 4 are available with the paper.

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

## Acknowledgements

This work was financially supported by the NIH (R01NS108151, R01GM121994 and R01-DK099023), the G. Harold & Leila Y. Mathers Foundation and the TESS Research Foundation (to D.N.W); and Wellcome Trust (210121/Z/18/Z) and BBSRC (BB/V007424/1) (to C.M.). D.B.S. was supported by the American Cancer Society Postdoctoral Fellowship (129844-PF-17-135-01-TBE) and Department of Defense Horizon Award (W81XWH-16-1-0153). We thank the following colleagues for helpful discussions: N. Coudray, R. Gonzalez Jr., M. Lopez Redondo, J.A. Mindell and E. Tajkhorshid. We are also grateful to colleagues at the Biophysics Colab, C. Grewer, R.M. Ryan and X. Wang, for commenting on the manuscript. We thank the staff at the NYU Cryo-EM Facility and the NYU Microscopy Core for assistance in grid screening and the Pacific Northwest Center for Cryo-EM in data collection. EM data processing used computing resources at the HPC Facility of NYULMC.

## Author contributions

J.J.M., J.S. and C.M. purified the proteins. J.J.M., J.C.S. and D.B.S. collected and analyzed the substrate-binding data. C.M. did all the cysteine PEGylation experiments. D.B.S collected and processed the cryo-EM images and built the atomic models. D.B.S and D.N.W. analyzed the structures. D.B.S., C.M. and D.N.W. wrote the manuscript. All authors participated in the discussion and manuscript editing. C.M. and D.N.W. supervised the research.

## Competing interests

The authors declare no competing interests.
