## [Peer Review File · Nature Communications]

REVIEWER COMMENTS

Reviewer #1 (Remarks to the Author):

The manuscript, Structural basis of ion – substrate coupling in the Na⁺-dependent dicarboxylate transporter VcINDY, is a thought provoking study about the biophysics of Na⁺-dependent transporters and would be of much interest to the membrane transport community.

VcINDY, is the prototype Na⁺-dependent dicarboxylate transporter from the divalent anion sodium symporter (DASS) family and is very well studied. The manuscript addresses the 'chicken or egg controversy' i.e. the effect of Na⁺ on the substrate binding site. The authors nicely spell out the induced fit mechanism, whereby Na⁺ binding causes a structural rearrangement to allow the substrate to bind. They resolve this induced fit mechanism in two ways:

- 1) Measuring the affinity of succinate in the absence and presence of Na⁺
- 2) Observing structural changes in the presence and absence of Na⁺.

Although I enjoyed this manuscript and I believe the authors provided substantial data to verify their hypothesis, there are a few points requiring clarification that will help the overall readability of the article.

- On page 4, the called the 'transport domain' the 'elevator domain' they should keep, it transport domain like the rest of manuscript
- The intrinsic tryptophan fluorescence quenching measurements are a classical binding assay and nicely preformed. The K_d measurement of 92.17 +/- 47.38 seems to have a high error of variation and a supplemental sentence to address this may be in order.
- K⁺ caused precipitation, was Li⁺ tried?
- Regardless, the assay clearly show Na⁺ is required for succinate binding.
- This is a general comment and I am not sure the best way to address it. Clearly, and demonstrated in SFig1 and Fig6, the transport cycle goes from an outward state where Na⁺ binding stabilizes the substrate binding site (the gist of the manuscript). After substrate binds the transporter transitions to the inward state where substrate is released then Na⁺ is released. It seems like a lot of this manuscript is inferring a lot of the outward state based on data from the inward sate of the transporter (all the data presented in the manuscript). There is no guarantees this is the case. In fact, there are many studies demonstrating the ordered reaction on the initial binding but stochastic release of Na⁺ and substrate on release. I feel there needs to be a clarification here so the reader can follow the mechanism of transport.

- The second main claim is a B-factor analysis of the structures based on the absence and presence of Na⁺. The author should clarify that a b-factor analysis is reliable at these resolutions (2.8-3.5).
- In general, figures could definitely be improved to clarify their claims. Figure 1 the b-factor analysis is lost in the figure. Or maybe have figures 1 and 2 side by side for direct comparison. Fig 2 C the movements are not observed well. Not sure its possible maybe a more direct looks at the Na sites?
- The site directed alkylation experiments were nicely performed and convincing.
- There was mention of transport assays done in proteoliposomes (pg 14) but it doesn't look like they were done in this manuscript to test mutants. Is that the case?
- There was mention the transporter use 3 Na⁺ to transport succinate but there are only 2 Na sites—this should be expanded upon to clarify.
- A better clarification of the loss of coordination at Na⁺ site might be useful. Describe coordination specifically and what is lost in the absence of Na⁺.
- Discussion should be more tight and related to the results. Not sure about the comparison with Glt... I will leave it to the authors discretion.

Overall, I enjoyed this manuscript and it should be published. I highlighted a few question that arose while reading it that I feel may strengthen the article, but overall it is very good and suitable for publication.

Reviewer #2 (Remarks to the Author):

Transporters work by an alternating access model where the membrane protein changes conformation displaying the substrate-binding site to the extra-cytosolic side and the cytosolic side of the membrane. The YCINDY sodium-dependent dicarboxylate transporter of the SLC13 family transports substrate in a sequential ordered mechanism where sodium binds first to the outward-facing state followed by succinate binding inducing a conformational change to an inward-facing state and release of the succinate and then the sodium. This paper reports the sodium-bound state without substrate and importantly, the apo-state adding to the previous sodium-succinate bound state filling a gap in our knowledge. A key finding is that sodium binding induces a conformational change in the protein to allow succinate binding in an induced fit model. Intrinsic fluorescence studies demonstrated that succinate does not binding to the transporter in the absence of sodium. Cysteine mutagenesis further revealed that key residues were occluded in the presence of sodium. The paper provides in exquisite molecular

detail (even identifying water molecules at the dimer interface) the nature of the sodium and succinate binding sites.

Minor Comments

- 1.(p.4) Structures of VcINDY (eg. CiNaS, CiNa) have been determined previously. The Introduction should more clearly describe these and how the new structures (CiNa in 300 mM Na, apoCi) described in this paper fill a knowledge gap in our understanding the mechanism of action of this transporter. What is known about Co, CoNa and CoNaS to complete the transport cycle?
- 2.(p.7) There are too many significant figures in the Kd values for succinate binding. Use 92 +/- 47 uM.
- 3.(p. 7) Were other members of the Hofmeister series tested other than K+? Li+? NH4+?
- 4.Does VcINDY bind and transport fumarate? What does the binding site reveal about specificity.
- 5.(p.10) What about the third Na+ site?
- 6.It is important to emphasis in the Discussion that the structures are all in the inward-facing state (Ci), so it is the order of sodium release that is being revealed. Place the structures in the context of the complete transport cycle: Co, CoNa, CoNaS, CiNaS, CiNa, Ci then back to Co as the empty carrier reorients.
- 7.(p 20) Of course, as stated it tis the transition between CoNaS to CiNaS and Ci to Co that are need to complete the transport cycle, nicely illustrated in Fig. 6.

Reviewer #3 (Remarks to the Author):

The authors have captured a sodium-binding and apo state of VcINDY, along with the previous reported sodium/substrate-binding state, they proposed an induced-fit model for this transporter. These results will lead a better understanding the transport cycle of this the divalent anion sodium symporter.

- 1.Only a few labels are present in the main figures that hinders the understanding for the readers.
- 2.A general overall structure with TM helices annotated in Figure 1 may lead a better understanding for the paper.
- 3.The residues that coordinate the sodium ion should compared with the Ci-Na+-S state to demonstrate the detail.

4. In the schematic model (Fig. 6), the relative positions of HPinb and TM10b should be the same for both protomers from this projection view, it is not accurate for the author to switch the two segments in the cartoon. Meanwhile, the labels of HPinb and TM10b are at the wrong position, the two short connected helices are HPinb and the third one in TM10b.

5. Model and map validation should be performed.

6. During the apo dataset processing, there are four good classes demonstrated, are there any differences? If No, they should be combined at earlier step to perform the 3D classification.

7. The author describes that "Substrate transport of VcINDY is driven by the inwardly-directed Na⁺ gradient, with dicarboxylate import coupled to the co-transport of three sodium ions (Supplementary Figs. 1a&b)". But the figures indicate that four sodium ions and two substrates are co-transported.

8. Supplementary Fig. 5. "Heterogeneous refin." and "Non-uniform refin." Should be corrected.

9. Counter level for the density map should be described.

REVIEWER COMMENTS

Reviewer #1 (Remarks to the Author):

The manuscript, Structural basis of ion – substrate coupling in the Na⁺-dependent dicarboxylate transporter VcINDY, is a thought provoking study about the biophysics of Na⁺-dependent transporters and would be of much interest to the membrane transport community.

VcINDY, is the prototype Na⁺-dependent dicarboxylate transporter from the divalent anion sodium symporter (DASS) family and is very well studied. The manuscript addresses the ‘chicken or egg controversy’ i.e. the effect of Na⁺ on the substrate binding site. The authors nicely spell out the induced fit mechanism, whereby Na⁺ binding causes a structural rearrangement to allow the substrate to bind. They resolve this induced fit mechanism in two ways:

- 1) Measuring the affinity of succinate in the absence and presence of Na⁺
- 2) Observing structural changes in the presence and absence of Na⁺.

Although I enjoyed this manuscript and I believe the authors provided substantial data to verify their hypothesis, there are a few points requiring clarification that will help the overall readability of the article.

- On page 4, the called the ‘transport domain’ the ‘elevator domain’ they should keep, it transport domain like the rest of manuscript

We thank the reviewer for catching our slip into the imprecise description of the transport domain. This has been corrected and the domain is now referred to as the transport domain throughout the revised manuscript.

- The intrinsic tryptophan fluorescence quenching measurements are a classical binding assay and nicely preformed. The K_d measurement of 92.17 +/- 47.38 seems to have a high error of variation and a supplemental sentence to address this may be in order.

We agree that the K_d measurement has a high error and we thank the reviewer for pointing this out. This is a result of the Tryptophan Fluorescence Quenching assay's low signal to noise due to physical and technical constraints on the experiments. Substrate binding produces a very small change in the total fluorescence, as the nearest Tryptophan is 7 Å from the substrate. Additionally, the protein concentration must be kept low to minimize inter-protein effects. Nevertheless, the experiment clearly demonstrates the sodium dependence of succinate binding.

- K⁺ caused precipitation, was Li⁺ tried?

We did not try Lithium, as this ion is known to drive substrate transport in VcINDY (Mancusso et al. Nature 2012, Mulligan et al. JGP 2014). Presumably therefore, Li⁺ can also occupy the cation binding sites to enable substrate binding.

- Regardless, the assay clearly show Na⁺ is required for succinate binding.

We thank the reviewer for their appreciation of our biochemical results validating Na⁺-dependent succinate binding.

- This is a general comment and I am not sure the best way to address it. Clearly, and demonstrated in SFig1 and Fig6, the transport cycle goes from an outward state where Na⁺ binding stabilizes the substrate binding site (the gist of the manuscript). After substrate binds the transporter transitions

to the inward state where substrate is released then Na⁺ is released. It seems like a lot of this manuscript is inferring a lot of the outward state based on data from the inward state of the transporter (all the data presented in the manuscript). There is no guarantee this is the case. In fact, there are many studies demonstrating the ordered reaction on the initial binding but stochastic release of Na⁺ and substrate on release. I feel there needs to be a clarification here so the reader can follow the mechanism of transport.

We thank the reviewers for raising this important point. The outward-facing state of other DASS co-transporters are known to also sequentially bind sodium the substrate (Pajor et al. *J. Membr. Biol.* 2013, Hill & Pajor. *J. Bacteriol.* 2005, Yao & Pajor. *Am. J. Physiol. Renal Physiol.* 2000), though the physical mechanisms underlying this process remain unknown. While it is easy to infer the pseudo-symmetric outward facing state operates using the same mechanism, this will need to be directly examined in future studies. We have clarified this point in the revised text.

- The second main claim is a B-factor analysis of the structures based on the absence and presence of Na⁺. The author should clarify that a b-factor analysis is reliable at these resolutions (2.8-3.5).

We thank the reviewer for raising this important point of comparing B-factors between structures of distinct resolutions. It is well known that normalizing is necessary when quantitative comparing B-factors between structures of distinct resolution (Ringo & Petsko. *Meth. Enzymol.* 1986, Carugo *Acta Cryst.* 2022). While we only qualitatively analyzed the intra-model differences in B-factors, we recognize this is unclear if the figures are plotted without normalization. We have therefore revised Figure 3 with panels c and d now illustrating normalized B-factors to make the differences readily apparent.

- In general, figures could definitely be improved to clarify their claims. Figure 1 the b-factor analysis is lost in the figure. Or maybe have figures 1 and 2 side by side for direct comparison.

We thank the reviewer for this suggestion to clarify the B-factor. We agree that a side-by-side illustration of the models is best for comparison. We have depicted this in Figure 3c-d. However, comparing B-factors between the C_i-Na and C_i-apo states did not reveal any additional differences. Therefore, we believe rearranging Figures 1 and 2 would provide little benefit while confusing the logical flow of the manuscript.

Fig 2 C the movements are not observed well. Not sure its possible maybe a more direct looks at the Na sites?

The reviewer raises a good point in illustrating the state dependent changes in the Na-binding regions. We have revised Figs 1f and 2c to clearly illustrate these changes and revised the text accordingly.

- The site directed alkylation experiments were nicely performed and convincing.

We are grateful the reviewer appreciates the value and clarity of these studies and thank them for their comment.

- There was mention of transport assays done in proteoliposomes (pg 14) but it doesn't look like they were done in this manuscript to test mutants. Is that the case?

We thank the reviewer for their highlighting this unclear point. We have not directly tested transport activity of the cysteine mutations. While transport is not required for our binding studies, our

previous work indicates any VcINDY mutants that express well and remain stable after purification are functional (Sampson, JBC, 2020). These points have been clarified in the revised manuscript.

- There was mention the transporter use 3 Na⁺ to transport succinate but there are only 2 Na sites—this should be expanded upon to clarify.

The reviewer raises a significant point which remains unknown in DASS co-transporter function, the binding sites of additional Na⁺ ions. Currently the location and effects of these ions within the structure remain unknown, and we have revised the text to make these points clear.

- A better clarification of the loss of coordination at Na⁺ site might be useful. Describe coordination specifically and what is lost in the absence of Na⁺.

We thank the reviewer for raising the importance of describing the exact Na⁺ interactions, and the changes in these interactions upon ion binding or release. We have revised Figs. 1f and 2c and the text to better clarify the exact ion-protein interactions and how these change through the reaction cycle's states. Particularly, it is important to clarify that the C_i-apo has no singular structure. Rather, the mobility of HP_{in} and TM10b creates an ensemble of structures (Fig. 6), as shown by our NMR-style analysis and supported by the site-directed alkylation experiments. To more accurately describe that Na⁺-binding captures a single structure from the C_i-apo state's structural ensemble, we have renamed this cooperative binding mechanism that we previously referred to as induced-fit to conformational selection. This point has been clarified in the revised manuscript.

- Discussion should be more tight and related to the results. Not sure about the comparison with Glt.... I will leave it to the authors discretion.

The reviewer raises an important factor in keeping the manuscript focused. We have significantly shortened this portion of the discussion to ensure the text is on-topic. We nevertheless believe a comparison to the Glt-family transporters is important as they are the only other elevator-type transporters that have been studied in similar detail. This is the first opportunity to examine differences in ion and substrate binding between these two families, and a short comparison is warranted.

Overall, I enjoyed this manuscript and it should be published. I highlighted a few question that arose while reading it that I feel may strengthen the article, but overall it is very good and suitable for publication.

Reviewer #2 (Remarks to the Author):

Transporters work by an alternating access model where the membrane protein changes conformation displaying the substrate-binding site to the extra-cytosolic side and the cytosolic side of the membrane. The YCINDY sodium-dependent dicarboxylate transporter of the SLC13 family transports substrate in a sequential ordered mechanism where sodium binds first to the outward-facing state followed by succinate binding inducing a conformational change to an inward-facing state and release of the succinate and then the sodium. This paper reports the sodium-bound state without substrate and importantly, the apo-state adding to the previous sodium-succinate bound state filling a gap in our knowledge. A key finding is that sodium binding induces a conformational

change in the protein to allow succinate binding in an induced fit model. Intrinsic fluorescence studies demonstrated that succinate does not binding to the transporter in the absence of sodium. Cysteine mutagenesis further revealed that key residues were occluded in the presence of sodium. The paper provides in exquisite molecular detail (even identifying water molecules at the dimer interface) the nature of the sodium and succinate binding sites.

Minor Comments

1.(p.4) Structures of VcINDY (eg. CiNaS, CiNa) have been determined previously. The Introduction should more clearly describe these and how the new structures (CiNa in 300 mM Na, apoCi) described in this paper fill a knowledge gap in our understanding the mechanism of action of this transporter.

We are grateful for the reviewer's note that the scientific justification for our structures is missing from the introduction, and only first discussed later in the manuscript. We have now revised the introduction to clearly explain the logic behind our experimental design.

What is known about Co, CoNa and CoNaS to complete the transport cycle?

The reviewer raises an important point regarding a complete description of the VcINDY reaction cycle. We have revised the manuscript to include what is known about the outward facing conformation and how our results relate to these states.

2.(p.7) There are too many significant figures in the K_d values for succinate binding. Use 92 +/- 47 uM.

The reviewer raises a good point regarding the apparent precision of our K_d measurement. Reviewing the raw data, all measurements had a minimum of three significant figures. Therefore, we have truncated the K_d to the same number of significant figures.

3.(p. 7) Were other members of the Hofmeister series tested other than K⁺? Li⁺? NH₄⁺?

We did not test Lithium as this ion is known to drive transport in VcINDY (Mancusso, Nature, 2012; Mulligan, JGP, 2014) and therefore can also occupy the cation binding sites to support succinate binding. We did not test NH₄⁺ as our aim was to ensure a fully empty cation binding sites while keeping the protein stable. While the ability of ammonium to drive transport has not been examined in VcINDY, the small difference in ionic radius between Na⁺ and NH₄⁺ risks it also occupying the binding site. We therefore chose choline, with its much larger ionic radius, to ensure fully apo cation binding sites.

4.Does VcINDY bind and transport fumarate? What does the binding site reveal about specificity.

The reviewer raises a fascinating point regarding VcINDY's substrates and the link between binding site and selectivity. While beyond the scope of this manuscript, the protein's substrate selectivity has been extensively studied elsewhere (Mancusso et al. Nature 2012, Mulligan et al. JGP 2014, Sampson et al. Biochem J. 2021)

5.(p.10) What about the third Na⁺ site?

We are grateful the reviewer's highlighting of an outstanding question in the DASS co-transporter field, the binding sites of additional co-transported cations. We have revised the manuscript to highlight the cryptic location and unknown effects of this ion. Further studies into the Na³ site are ongoing and will be the subject of future publications.

6. It is important to emphasize in the Discussion that the structures are all in the inward-facing state (Ci), so it is the order of sodium release that is being revealed. Place the structures in the context of the complete transport cycle: Co, CoNa, CoNaS, CiNaS, CiNa, Ci then back to Co as the empty carrier reorients.

We appreciate the reviewer's point of discussing our observed link between protein structure and sequential binding within the context of the transporter's entire reaction cycle. We have clarified this point in the revised manuscript.

7.(p 20) Of course, as stated it is the transition between CoNaS to CiNaS and Ci to Co that are needed to complete the transport cycle, nicely illustrated in Fig. 6.

We kindly thank the reviewer for appreciating the mechanistic points of the VcINDY reaction cycle clarified by Figure 6's schematic.

Reviewer #3 (Remarks to the Author):

The authors have captured a sodium-binding and apo state of VcINDY, along with the previously reported sodium/substrate-binding state, they proposed an induced-fit model for this transporter. These results will lead to a better understanding of the transport cycle of this divalent anion sodium symporter.

1. Only a few labels are present in the main figures that hinders the understanding for the readers.
2. A general overall structure with TM helices annotated in Figure 1 may lead to a better understanding for the paper.

We are grateful for the reviewer's point that the main text figures would be much clearer with important features labeled. We have modified and relabeled these figures in the revised submission to clarify the relevant structural elements. We have also added an additional figure to the supplement (Supplementary figure 1e in the revised manuscript) to help the reader place features within the context of the larger transport domain.

3. The residues that coordinate the sodium ion should be compared with the Ci-Na⁺-S state to demonstrate the detail.

We thank the reviewer for raising the importance of describing the exact Na⁺ interactions, and the changes in these interactions upon ion binding or release. We have revised Figs 1f and 2c to illustrate these points, and we have clarified in the text the exact ion-protein interactions and how these change through the reaction cycle's states.

4. In the schematic model (Fig. 6), the relative positions of HPinb and TM10b should be the same for both protomers from this projection view, it is not accurate for the author to switch the two segments in the cartoon. Meanwhile, the labels of HPinb and TM10b are at the wrong position, the two short connected helices are HPinb and the third one is TM10b.

We thank the reviewer for pointing out the error in mislabelling HP_{in}b and TM10b in the reaction cycle's schematic model, which are now relabelled. We have modified the figure to better clarify structural features of each protomer and corrected the labelling.

5. Model and map validation should be performed.

We thank the reviewer for pointing out that map and model validation statistics are critical to evaluating the quality of the results. Maps and models were individually evaluated using standard statistics of the field (Supplementary Fig. 3c, 6b, and Table S1). Map-to-model refinement was validated using the state-of-the-art FSC statistic (Table S1) (Afonine et al. Acta Cryst. D. 2018, Neumann et al. Structure 2018)

6. During the apo dataset processing, there are four good classes demonstrated, are there any differences? If No, they should be combined at earlier step to perform the 3D classification.

The reviewer raises an important point regarding the C1 maps generated while processing the VcINDY-choline dataset. Combining the particles resulted in a poorer quality map and lower resolution. Notably, the C1 maps represent varying structures of the C₁-apo structural ensemble. We have included a new figure in the supplement (Supplementary Fig. 7) illustrating the differences between the protomers of each map.

7. The author describes that "Substrate transport of VcINDY is driven by the inwardly-directed Na⁺ gradient, with dicarboxylate import coupled to the co-transport of three sodium ions (Supplementary Figs. 1a&b)". But the figures indicate that four sodium ions and two substrates are co-transported.

The reviewer raises two important points regarding the clarity of the schematic figure, and we are appreciative of the note. The 3:1 Na⁺ to substrate stoichiometry is for a protomer, and the illustrations in Fig. 6 and Supplementary Fig. 1b are for a dimer, which is the physiologically-relevant oligomeric state. In each protomer, the binding site for VcINDY's third Na⁺ ion is unknown and therefore only Na1 and Na2 are shown for clarity. Additionally, the two transport domains are shown symmetrically for simplicity only. We are actively examining the coupling between transport domains in VcINDY by single molecule FRET, current structural and functional data indicate the transport is not coupled between protomers. These points have been clarified in the figure legend and text.

8. Supplementary Fig. 5. "Heterogeneous refin." and "Non-uniform refin." Should be corrected.

We thank the reviewer for pointing out these non-standard abbreviations and have spelled-out both terms in the revised supplementary figure.

9. Counter level for the density map should be described.

We thank the reviewer for pointing this out and have included contour levels for each map in the revised manuscript.

REVIEWERS' COMMENTS

Reviewer #1 (Remarks to the Author):

All comments have been addressed to satisfy ton.

Reviewer #2 (Remarks to the Author):

The authors have addressed the comments made in my review in this revised manuscript. The new structures presented in exquisite molecular detail fill vital gaps in our knowledge of the mechanism of ion-coupled transporters.

Reviewer #3 (Remarks to the Author):

Most issues are well addressed except for the model and map cross validation. The author mentions that they have performed the Individual validations of both map and model but it is not cross validations between the models and maps.

Response to reviewers' comments

Reviewer #1 (Remarks to the Author):

>All comments have been addressed to satisfy ton.

Reviewer #2 (Remarks to the Author):

*>The authors have addressed the comments made in my review in this revised manuscript. The new
>structures presented in exquisite molecular detail fill vital gaps in our knowledge of the mechanism of ion-
>coupled transporters.*

Reviewer #3 (Remarks to the Author):

*>Most issues are well addressed except for the model and map cross validation. The author mentions that
>they have performed the Individual validations of both map and model but it is not cross validations
>between the models and maps.*

Indeed, we performed cross validations between the models and the maps. The values are listed in the last line in Supplementary Table 1 as Model resolution[†] (Å), which is defined as “Resolution cutoff at which the model and sharpened map Fourier coefficients reach 0.143.” Perhaps the reviewer missed it.